# U-CAN-seq: A Universal Competition Assay by Nanopore Sequencing

**DOI:** 10.3390/v16040636

**Published:** 2024-04-19

**Authors:** Jennifer Diaz, John Sears, Che-Kang Chang, Jane Burdick, Isabella Law, Wes Sanders, Colton Linnertz, Paul Sylvester, Nathaniel Moorman, Martin T. Ferris, Mark T. Heise

**Affiliations:** 1Department of Microbiology and Immunology, University of North Carolina at Chapel Hill, Chapel Hill, NC 27514, USA; jloome@live.unc.edu (J.D.);; 2Department of Genetics, University of North Carolina at Chapel Hill, Chapel Hill, NC 27514, USA; 3The Rapidly Emerging Antiviral Drug Development Initiative (READDI), Chapel Hill, NC 275114, USA

**Keywords:** competition assay, nanopore sequencing, alphavirus, RNA virus, viral fitness

## Abstract

RNA viruses quickly evolve subtle genotypic changes that can have major impacts on viral fitness and host range, with potential consequences for human health. It is therefore important to understand the evolutionary fitness of novel viral variants relative to well-studied genotypes of epidemic viruses. Competition assays are an effective and rigorous system with which to assess the relative fitness of viral genotypes. However, it is challenging to quickly and cheaply distinguish and quantify fitness differences between very similar viral genotypes. Here, we describe a protocol for using reverse transcription PCR in combination with commercial nanopore sequencing services to perform competition assays on untagged RNA viruses. Our assay, called the Universal Competition Assay by Nanopore Sequencing (U-CAN-seq), is relatively cheap and highly sensitive. We used a well-studied N24A mutation in the chikungunya virus (CHIKV) nsp3 gene to confirm that we could detect a competitive disadvantage using U-CAN-seq. We also used this approach to show that mutations to the CHIKV 5′ conserved sequence element that disrupt sequence but not structure did not affect the fitness of CHIKV. However, similar mutations to an adjacent CHIKV stem loop (SL3) did cause a fitness disadvantage compared to wild-type CHIKV, suggesting that structure-independent, primary sequence determinants in this loop play an important role in CHIKV biology. Our novel findings illustrate the utility of the U-CAN-seq competition assay.

## 1. Introduction

RNA viruses, which generally lack proofreading activity, evolve quickly by acquiring mutations in their small, labile genomes that shape multiple aspects of their biology, including viral fitness, immune escape, viral transmissibility, and host range. These changes can impact viral pathogenesis or even shape the course of epidemics [1,2]. As such epidemics unfold, researchers often try to understand how novel genotypic variants will affect viral spread, especially in the context of resistance to antiviral drugs or escape from vaccine-induced immunity. However, small changes in RNA virus genomes can have major effects on viral fitness while having no detectable growth defects in vitro. For instance, a mutation in the SARS-CoV-2 RNA-dependent RNA polymerase that conferred reduced sensitivity to the antiviral drug remdesivir did not affect virus replication in growth curves, but it did confer a competitive disadvantage compared to WT virus in a competition assay [3]. The same pattern was shown with zidovudine resistance mutations in HIV-1 [4], nevirapine resistance mutations in HIV-1 [5], and baloxivir resistance mutations in influenza A virus [6]. All of these resistance mutations caused no overt growth defects but did show disadvantages in the context of competition between viral genotypes, informing work to prevent the spread of drug resistant viruses. 

Competition assays provide a method to stringently and sensitively measure the relative fitness of two virus genotypes [7,8,9,10]. During competition assays, two virus genotypes infect the same cell culture well or animal tissue, and the ratio of virus genotypes is measured in the inoculum and in the post-infection samples. As viruses compete to infect available cells and produce progeny genomes, small differences in virus replication rates become amplified over multiple rounds of replication. This allows researchers to measure subtle changes in replication that can have large effects on pathogenesis and epidemics. Competition assays require quantification of the relative abundance of two very similar genotypes of virus, often differing by only one or a few nucleotides, and this poses technical challenges. Some researchers use viruses encoding fluorescent tags, which allow detection of the genotype ratios via flow cytometry [9], or genetic tags, which allow detection of the virus genotypes by standard quantitative PCR [7]. However, tagging the viruses may confound studies by introducing small, or even large, advantages or disadvantages in replication [11,12,13,14]. RNA sequencing can detect untagged viruses [10], but it can be cost-prohibitive [15,16], require complex technical processing pipelines [17], and, in some cases, may come with multi-week delays between running experiments and getting sequencing results.

Commercial nanopore sequencing services allow the complete sequencing of plasmid or linear amplicon sequences at a relatively low cost with rapid data turnaround. We have leveraged commercial nanopore sequencing to develop a cost-effective competition assay for highly similar, untagged genotypes of RNA viruses with rapid data turn-around times. Based on this, we developed a highly accessible competition assay protocol that combines reverse transcription (RT)-PCR with commercial nanopore sequencing, which we call Universal Competition Assay by Nanopore Sequencing, or U-CAN-seq. We then used several previously described CHIKV mutants to demonstrate that the assay can rapidly and quantitatively detect fitness differences between two closely related, untagged viruses. Lastly, we used U-CAN-seq to detect a previously unrecognized fitness difference due to mutations in CHIKV RNA structures. 

## 2. Materials and Methods

### 2.1. Cell Culture and Viruses

Vero81 cells (ATCC CCL-81, Manassas, VA, USA) were cultured in DMEM (Gibco, Waltham, MA, USA) + 10% fetal bovine serum (FBS, R&D Systems, Minneapolis, MN, USA) + 1% L-glutamine (Gibco, Waltham, MA, USA) at 37 °C in 5% CO_2_. All viruses were generated from plasmid-encoded infectious clones by in vitro transcription (mMessage Machine Sp6 kit, Ambion, Austin, TX, USA) to produce viral RNA, which was electroporated into BHK21 cells (ATCC CCL-10, Manassas, VA, USA). The virus was recovered from cell supernatants at 24–28 h post infection. All genotypes of virus described were on the 181/25 CHIKV background.

### 2.2. Growth Curves

We performed growth curves by infecting Vero81 cells with each virus at a multiplicity of infection (MOI) = 0.01 in 400 µL of medium in 6-well plates, rocking every 15 min for 1 h at 37 °C. The cells were then washed three times with Dulbecco’s phosphate-buffered saline (DPBS, Gibco, Waltham, MA, USA) to remove unbound input virus, and 3 mL of medium was added to each well prior to incubation at 37 °C. We harvested 100 µL of supernatant from each well and replaced it with fresh medium at 1, 8, 16, and 24 h post infection (hpi), and quantified viral titers by plaque assay as described previously [18]. Briefly, supernatants were serially diluted 1:10 six times and plated in duplicate on 90% confluent Vero81 cells, the plates were rocked every 15 min for one hour, and 2 mL of 1.25% carboxymethylcellulose (Sigma-Aldrich, St. Louis, MO, USA) overlay in MEM alpha (Gibco, Waltham, MA, USA) was added. Plaques were allowed to form for 3 days before plates were fixed by adding 2 mL of 4% paraformaldehyde (Thermo Scientific, Waltham, MA, USA) washing, and staining in 0.25% crystal violet (Fisher Chemical, Pittsburgh, PA, USA). 

### 2.3. Competition Assays

The full U-CAN-seq protocol is published at protocols.io (https://www.protocols.io/private/F8AEFAE4D19711EE9D470A58A9FEAC02, accessed on 27 February 2024). Briefly, we performed competition assays by infecting Vero81 cells at an MOI = 0.01 in 400 µL in six-well plates, rocking every 15 min for 1 h, and adding 3 mL of medium as described above. We infected the cells with each virus genotype alone, along with mixes of two genotypes (a wild-type and a mutant) in 1:1, 5:1, and 10:1 ratios, infecting five wells each. 300 µL of each inoculum was mixed with 900 µL of Trizol-LS (Invitrogen, Carlsbad, CA, USA). The infected cells were incubated for 24 h at 37 °C, at which point, we harvested 300 µL of supernatant from each well, which was then mixed with 900 µL of Trizol-LS to isolate RNA. 

We isolated RNA from each sample (3 replicate inoculum samples and 5 replicate output supernatant samples per genotype ratio) according to the Trizol-LS Reagent (Invitrogen, Carlsbad, CA, USA) manufacturer’s instructions. We used a gene-specific reverse primer with the Superscript III first-strand synthesis kit (Invitrogen, Carlsbad, CA, USA) to generate cDNA (Table 1). Gene-specific PCR primers were used to amplify the region around each region of interest for 35 PCR cycles (primers listed in Table 1). Amplification of the target, and lack of amplification of a mock sample, was confirmed by running the samples on an agarose gel (Appendix A). We used the DNA Clean & Concentrator-5 kit (Zymo Research Corporation, Irvine, CA, USA) to purify DNA from each sample, eluting in 12 µL. We performed one PCR replicate per sample.

We submitted each amplicon to the Plasmidsaurus amplicon sequencing service in Louisville, KY, USA, which performs Oxford Nanopore sequencing using the v14 library prep chemistry and R10.4.1 flow cells. FASTQ files containing each raw sequencing read were returned. The FASTQ files from our amplicon sequencing were analyzed using simple command-line prompts that can be used in any Unix-based system through a command line interface (CLI). Code to perform the analysis and create an output csv file summarizing the data can be found in Appendix A. We used the Unix ‘grep’ command, a utility designed to search for matching strings in text files, to identify and count specific k-mers (Table 1) (10–40 nucleotides containing the site of interest, that are unique in the CHIKV genome otherwise) in our sequencing data. Note that this method works better than standard single nucleotide polymorphism (SNP) calling if at least two nucleotides differ between the two genotypes. In this way, we can avoid misalignments and misestimates of abundances of genotypes due to sequencing errors near our relevant mutant sites. We counted the number of occurrences of each k-mer in each FASTQ file, and we calculated the proportion of each k-mer compared to the total number of wild-type (WT) and mutant k-mers counted for that file. Total k-mer occurrences in the sequencing files varied widely among samples, from 4 to 1712 (Appendix A). We used one-way ANOVA followed by relevant pre-planned contrasts for pairwise comparisons to compare the mean genotype frequencies of the rarer genotypes of the supernatants to the relevant inocula for each ratio of genotypes. We used Sidak’s multiple comparisons test to control for multiple comparisons. We determined results as significant if *p* < 0.05. (* *p* < 0.05, ** *p* < 0.01, *** *p* < 0.001, **** *p* < 0.0001). We expect that genotypes that have no fitness difference compared to the wild-type should have equal genotype ratios in the inocula and in the post-infection samples, while genotypes that have a competitive advantage or disadvantage have higher or lower proportions, respectively, in the post-infection samples than in the inocula. 

## 3. Results

### 3.1. U-CAN-seq Detects a Previously Described Growth Defect in the CHIKV nsp3 N24A Genotype

To validate the U-CAN-seq protocol, we first analyzed the relative fitness of a chikungunya virus (CHIKV) mutant with a known growth defect compared to WT CHIKV. For this proof-of-concept experiment, we chose a mutation within the nsP3 macrodomain (N24A). The N24A mutation disrupts the ribosylhydrolase activity of the nsP3 macrodomain, which results in attenuated growth [19]. We first confirmed its growth defect by performing a multi-step growth curve of WT and N24A CHIKV on Vero81 cells at an MOI of 0.01 (Figure 1). By 24 h post infection (hpi), the N24A genotype had titers approximately 10-fold lower than the WT. Next, we performed a competition assay to determine if N24A had a fitness advantage or disadvantage compared to the WT virus, with the expectation that we would detect a fitness disadvantage compared to the WT. We infected Vero81 cells at a ratio of 1:1, 5:1, and 10:1 WT:N24A virus and isolated RNA from the inoculum and from cell supernatants at 24 hpi. We used RT-PCR to amplify the target region in nsP3 and submitted the amplicons for nanopore sequencing. 

We successfully amplified sequences from samples with RNA concentrations as low as 6 ng/µL, and although the inocula had less robust amplification on average than the supernatants of infected cells (Appendix A), our sequence of interest was consistently amplified from both supernatants and the inoculum. The N24A mutant had a lower genotype frequency in the supernatant of infected cells than it did in the inoculum, confirming that it is less fit than the WT virus (Figure 1) and confirming that our assay can detect a known growth defect. Of note, the assay was also robust even when the input inoculum was inexact. For example, for the intended 1:1 WT:N24A input inoculum, we calculated an actual genotype ratio of 1:9.1. This deviation from the desired input ratios was most likely due to titering the WT and N24A viruses on separate days, leading to variation. We therefore recommend titering viruses of interest side-by-side before performing U-CAN-seq to ensure accurate titers. Regardless, N24A had a reduced allele frequency at output compared to the input. Because we statistically compare input to output rather than to an expected ratio, our assay is robust and interpretable even if input ratios vary from the intended ratios.

### 3.2. U-CAN-seq Can Detect Previously Unrecognized Fitness Differences Due to Mutations in an RNA Structure 

Next, we used U-CAN-seq to analyze fitness effects of a panel of previously described mutants in the 5′ end of the CHIKV genome that were used to characterize the role of RNA secondary structure in CHIKV replication [18]. We had previously shown that disruption of secondary structure in the two stem loops that comprise the alphavirus 5′ conserved sequence element (5′ CSE), as well as a stem loop immediately prior to the 5′ CSE, SL3, resulted in attenuation of CHIKV replication [18]. Structure rescue mutations in SL3 (scrSL3) and the 5′CSE (scr5′ CSE) that restored the secondary structure while changing the primary sequence had no impact on CHIKV replication compared to the wild-type virus, thereby demonstrating the importance of RNA secondary structure in these stem loops for viral replication [18]. To test whether these structure rescue mutants had more subtle fitness defects, we performed a U-CAN-seq assay. We first confirmed that scrSL3 and scr5′ CSE had no detectable effect on viral growth by testing the viruses head-to-head in multistep growth curves in Vero81 cells (Figure 2). When compared to WT CHIKV in our competition assay, the scr5′CSE virus showed no differences in fitness (Figure 3B). Because we have previously shown that disruption of the predicted secondary structure of the 5′ CSE reduces viral replication [18], our results here further confirm that the impact of the 5′ CSE on viral replication is mediated through RNA secondary structure. However, the scrSL3 genotype made up a lower proportion of post-infection sequences than of the inoculum sequences, indicating that scrSL3 has a competitive disadvantage compared to the wild-type virus (Figure 3A). Therefore, these studies indicate that although the secondary structure of SL3 is clearly important for viral replication [18], the primary sequence in this region also plays a role in viral fitness. To confirm our initial results, we performed competition assays again on scrSL3 and scr5′ CSE, this time including an input ratio in which the mutant was in the majority (WT:mutant 51:5) (Figure 3C,D). As in our initial experiment, we detected a competitive disadvantage in scrSL3 but not in scr5′ CSE compared to the WT. This demonstrates U-CAN-seq is capable of detecting fitness differences between viral genotypes at a variety of input genotype ratios. 

## 4. Discussion

Competition assays are an important virology tool for sensitively assessing viral fitness, but many of the existing systems for performing competition assays rely either on tagged viruses or high throughput RNA sequencing approaches, which increase the required time and costs of these assays. Furthermore, while tagged viruses have been highly useful for testing virus fitness, the potential impact of these tags on viral fitness often requires the inclusion of additional controls, increasing the time and cost commitment associated with these studies. Therefore, the protocol for a Universal Competition Assay by Nanopore sequencing (U-CAN-seq) is an important improvement upon previous protocols by providing a relatively inexpensive and powerful tool to perform competition assays on untagged viruses. U-CAN-seq expands the new accessibility of high-throughput DNA sequencing to a problem that traditionally is solved using RNA sequencing, which remains expensive despite its steady decrease in price. 

U-CAN-seq has a current cost of about $950 per experiment, for 33 samples per experiment (Table 2) after RNA extraction is performed, for researchers in the continental United States. The two largest current expenses are the reverse transcriptase and sample sequencing. It may be possible to reduce expenses by using a less costly and less processive reverse transcriptase enzyme, although we validated our protocol using only the listed reagents. Importantly, U-CAN-seq also expands the accessibility of sequencing-based competition assays to researchers at small institutions without bioinformatics collaborators because it requires only a single line of code, usable on common operating systems for easy analysis. We hope that our technique will allow even undergraduate-based research laboratories to carry out competition assays. 

As noted above, U-CAN-seq allows the researcher to investigate the relative fitness of two RNA viruses without adding tags, eliminating both the cost of creating a tagged infectious clone and any artificial fitness costs that a tag might introduce. Introducing fluorescent tags into small RNA genomes can allow for easy and cheap quantification of two viruses in a mixed population, but can also come at notable fitness costs, with many reporter systems facing issues of unstable transgenes [11,12,13,14,20]. These fitness costs may be due to disruptions in genome packaging, the diversion of resources to produce non-advantageous fluorescent proteins, or other unintended disruptions. It is also possible to insert short sequence tags into RNA viruses to allow for quantification of two genotypes by qPCR. Even if these tags do not disrupt the coding sequence or fall within untranslated regions, one must be careful about disrupting potentially important RNA secondary structures [18,21,22,23,24] or other RNA regulatory elements that might also impact viral fitness. U-CAN-seq avoids these potential issues by eliminating the need for tagged viruses. 

Nanopore sequencing is notably error prone, with some single-nucleotide errors, as well as high rates of called insertions and deletions [25,26,27]. This makes performing local alignments somewhat challenging. Additionally, RNA viruses have relatively high mutation rates that can contribute to error in ratio assessment. For instance, in our datasets, our regions of interest differ from the exact k-mer sequences by one nucleotide in 0–2% of reads (Appendix A). This suggests that despite the high error rate of nanopore sequencing, errors are introduced to our small sequence of interest at quite a low rate. Additionally, our method of searching strings of nucleotides for exact matches eliminates erroneous reads from analysis, increasing the specificity of the data. Of course, searching for perfect matches also eliminates any reads with local errors, effectively reducing the read depth of the assay by 0–2% compared to assays that tolerate single-nucleotide mismatches. However, we believe this reduction in depth is effectively countered by our increased precision in assessment of relevant genotypes.

We assessed within-treatment-group standard deviations in the mutant genotype frequency across experimental replicates, and we found that these values were relatively small (median standard deviation in input inoculum 0.0358, range 0.0011–0.1178; median in harvested supernatants 0.0262, range 0.0043–0.1128). The estimated variation shrinks moderately when we consider groups where all samples had at least 50 reads including either the WT or the mutant k-mer (median standard deviation in input inoculum 0.0214 range 0.0008–0.1071, or in harvested supernatants median 0.024, range = 0.0043–0.1128). We used these values to estimate the relative frequency changes we would be powered to detect using U-CAN-seq (Table 3). In general, even at low (e.g., 3) within-group sample sizes (where a sample is RNA from the supernatant of a single well of infected cells), we can detect genotype frequency changes of ~8–12%, and this number shrinks to ~4% at sample sizes of 10. The high power of this protocol makes it useful for teasing out small differences in fitness without extensive passaging of virus. We found that the largest source of variation in calculated genotype frequencies in this assay is variation in read depth during nanopore sequencing, with low read depth resulting in less precise genotype frequency measurements. We therefore recommend aiming for read depths of >100 reads in our target region. 

Due to the high power of the assay, researchers can deploy U-CAN-seq to study the relative fitness of RNA viruses with small fitness differences and minimal sequence differences. In proof-of-concept studies using U-CAN-seq, we demonstrated that in CHIKV, the nsp3 N24A mutation confers a competitive disadvantage in cell culture compared to the WT. This finding is consistent with previously published data showing that in Sindbis virus, the nsp3 N24A mutation slightly reduces viral growth in cell culture [28,29]. It is also consistent with studies demonstrating that the N24A mutation reduces mono-ADP-ribosylhydrolase activity [19], inhibition of which decreases viral replication [30]. We also used U-CAN-seq to investigate more rigorously the sequence requirements of two RNA structures, showing that in Vero81 cells, disruption to the sequence but not the structure of the 5′ CSE in CHIKV does not affect viral fitness, but that disruption of the sequence but not the structure in SL3 does have a fitness cost (Figure 3). This novel finding illustrates the importance of using rigorous tests such as competition assays to understand the complex evolutionary pressures on RNA virus sequence and structure, as well as the utility of our novel approach to competition assays. 

We have demonstrated the effectiveness of U-CAN-seq to analyze the relative fitness of two genotypes of RNA virus in cell culture. The U-Can-seq protocol remains to be tested in animal models and other non-cell culture systems, but there is no conceptual reason why the protocol could not be adapted beyond cell culture. The technique is specifically valuable to analyze genotypes that differ at one specific locus and can be applied to any RNA virus. However, U-CAN-seq may be especially powerful when used to study alphaviruses because cells infected with alphaviruses become resistant to further infection by alphaviruses within 15 min of initiation of the first infection [31,32,33,34,35]. This effect, called superinfection exclusion, does not occur in all RNA viruses and causes small differences in replication speed to confer a major competitive advantage on the faster viral genotype, increasing the effect size in a competition assay. 

U-CAN-seq may be less useful for analysis of genotypes that differ at many locations across the genome. It is essential that the two genotypes of interest do not differ at the primer binding site in order to not introduce excessive bias in the PCR step. This is particularly essential because, with our method, it is not possible to determine how many individual genomes were amplified and sequenced, so avoidance of bias is crucial. U-CAN-seq may also be less useful in the study of insertions or deletions, which could cause preferential amplification of the shorter genotype. PCR bias may still occur, causing preferential amplification of one genotype over the other. However, the inoculum and output samples are subject to the same PCR bias, and thus any differences in genotype ratio between inoculum and output should represent true biological changes. Despite these limitations, U-CAN-seq is a powerful tool that can be used by any lab, regardless of institutional sequencing facilities, to understand the relative fitness of RNA viruses.

## Figures and Tables

**Figure 1 viruses-16-00636-f001:**
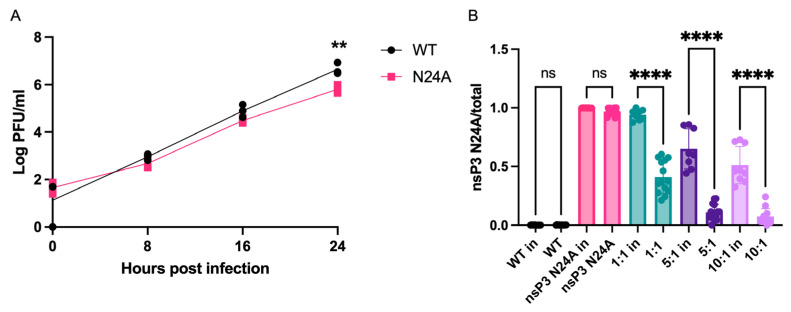
U-CAN-seq can reliably detect a fitness defect caused by the chikungunya virus (CHIKV) nsp3 N24A genotype. (**A**) Vero81 cells were infected with wild-type (WT) or nsp3 N24A CHIKV and samples were taken at 0, 8, 16, and 24 h post infection (hpi). Titers were measured by plaque assay. One representative experiment of three independent experiments shown. (**B**) Vero81 wells were infected at a multiplicity of infection of 0.01 with CHIKV WT (black), nsp3 N24A (pink), or 1:1 (turquoise), 5:1 (purple), or 10:1 (light purple) ratios of WT:N24A. RNA was extracted from the inocula and from 24 hpi cell supernatants, and reverse transcription (RT)-PCR was used to generate amplicons for nanopore sequencing. The raw sequencing files were searched for genotype-specific sequences and the proportions of the two genotypes in each sample were calculated. Combined results of three independent experiments shown. ns, not significant, ** *p* < 0.01, *****p* < 0.0001.

**Figure 2 viruses-16-00636-f002:**
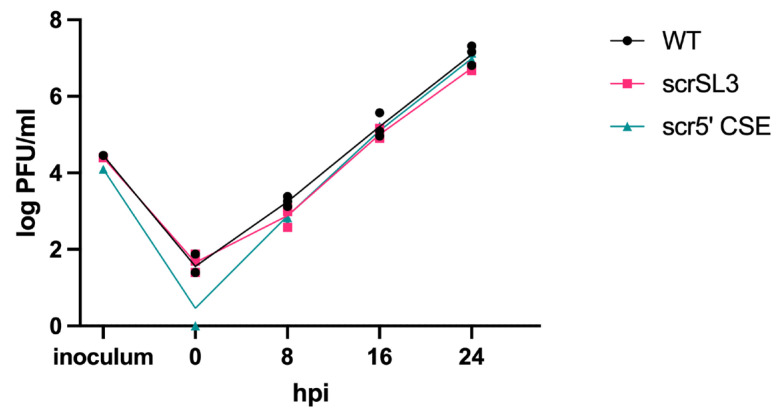
Mutations that impact the RNA sequence but not the predicted structure of the stem loop 3 (SL3) and the 5′ conserved sequence element (5′ CSE) have no detectable impact on the titer. Vero81 cells were infected with the WT, scrSL3, or scr5′ CSE CHIKV virus, and samples were taken at 0, 8, 16, and 24 h post infection. Titers were measured by plaque assay. One representative experiment of three independent experiments shown.

**Figure 3 viruses-16-00636-f003:**
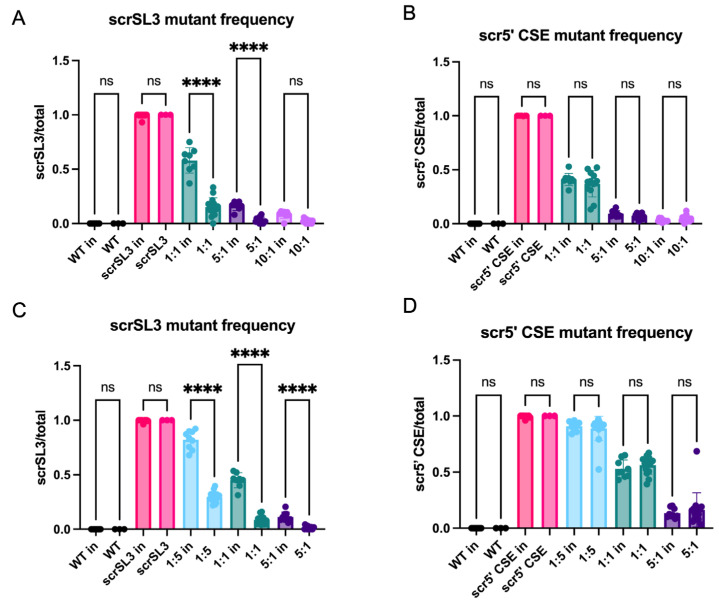
U-CAN-seq detects a fitness defect in the scrSL3 mutant but not the scr5′ CSE mutant. Vero81 wells were infected at a multiplicity of infection of 0.01 with CHIKV WT (black), scrSL3 (pink) (**A**,**C**), or scr5′ CSE (pink) (**B**,**D**) or with the indicated ratios of WT–mutant. (**A**,**B**) WT–mutant ratios in inocula were 1:1 (turquoise), 5:1 (purple), and 10:1 (light purple). (**C**,**D**) WT–mutant ratios in inocula were 1:5 (blue), 1:1 (turquoise), and 5:1 (purple). Inocula were sampled, and supernatants were sampled at 24 h post infection. RNA was extracted, and RT-PCR was used to generate amplicons for nanopore sequencing. The raw sequencing files were searched for genotype-specific sequences and the proportions of the two genotypes in each sample were calculated. Combined results of three independent experiments shown. ns, not significant, **** *p* <0.0001.

**Table 1 viruses-16-00636-t001:** Primers and sequences.

	nsp3 N24A	scrSL3	scr5′ CSE
Mutation location (base pairs in CHIKV genome)	4145–4147	91–145	175–217
RT primer	CGCAGTCTATGGAGATGTGCTCATC	CATGAACGGGGTTGTGTCGAACCC	CATGAACGGGGTTGTGTCGAACCC
PCR forward primer	GGTACTGGTGGGCGACTTGACTAATCCGCCC	GCAAAGCAAGAGATTAATAACCC	GCAAAGCAAGAGATTAATAACCC
PCR reverse primer	CGCAGTCTATGGAGATGTGCTCATC	CATGAACGGGGTTGTGTCGAACCC	CATGAACGGGGTTGTGTCGAACCC
WT k-mer	AGAGTGCGTGGTTAACGCCGCCAAC	ACGTGGACATAGACGCTGACAGCGCCTTTTT	ATCGAATGACCATGCTAATGCTAGAGCGTTCTCGCATCTAGCC
Mutant k-mer	AGAGTGCGTGGTTAACGCCGCCGCC	ATGTAGACATAGACGCTGACAGCGCCTTTCT	GTCGAATGACCATGCCAACGCCAGAGCGTTTTCGCATCTGGCT

The primers listed were used for reverse transcription and PCR. The k-mers listed were used with a Unix ‘grep’ search to search the raw fastq sequencing results files for genotype-specific sequences (k-mers, where k = length of the relevant sequence string).

**Table 2 viruses-16-00636-t002:** Costs associated with U-CAN-seq protocol.

Step	Reagent	Supplier	Product Number	Reactions	Cost of Kit	Reactions/Kit	Cost
RT	Superscript III first-strand synthesis kit	Invitrogen, Carlsbad, CA, USA	18080051	33	558	50	368.28
	Reverse primer	Invitrogen, Carlsbad, CA, USA			14	25 nmol	14
	PCR tubes	Greiner, Kremsmünster, Austria	07-000-717	33	254.5	960	8.75
PCR (20 µL reaction)	Phusion kit	NEB, Ipswich, MA, USA	M0530L	33	480	1250	12.67
	Forward primer	Invitrogen, Carlsbad, CA, USA			14	25 nmol	14
	dNTPs	NEB, Ipswich, MA, USA	N0447L	33	274	2000	4.52
	PCR tubes	Greiner, Kremsmünster, Austria	07-000-717	33	254.5	960	8.75
Clean up	Zymo DNA Clean & Concentrator-5	Zymo, Irvine, CA, USA	D4004	33	323	200	53.30
Sequencing	Amplicon sequencing	Plasmidsaurus, Louisville, KY, USA		33	15	1	495
						SUM	979.27

**Table 3 viruses-16-00636-t003:** Power estimates: Expected detectable effects testing for an alpha = 0.05, beta = 0.9.

	St.Dev = 0.0358	St.Dev = 0.0214
*n* = 3	11.73%	7.97%
*n* = 5	7.67%	5.21%
*n* = 10	5.01%	3.4%

Using two different estimates of the standard deviation of genotype frequencies of our assay (the highest and lowest median standard deviation estimates), we calculated the minimum effect size our assay has the power to detect for a given sample size, assuming an alpha (likelihood of incorrectly calculating a statistically significant difference between two groups) of 0.05 and a beta (probability of incorrectly failing to identify a statistically significant difference) of 0.9.

## Data Availability

The raw data supporting the conclusions of this article will be made available by the authors on request.

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
