# Peer review of "U-CAN-seq: A Universal Competition Assay by Nanopore Sequencing"

_viruses, 2024, doi:10.3390/v16040636_

Round 1

Reviewer 1 Report

Comments and Suggestions for Authors

The MS by Diaz et al. discusses interesting and important results both from a basic but also from an applied science point of view. Competition assays are important to evaluate the effects of different mutations on the fitness of the affected virus strains.

The MS is generally well-written and usually easy to follow. There are some repetitions though, one can argue if these aid the understanding of the MS or are unnecessary.

The only major concern of this reviewer regarding the MS is the use of repeated experiments. How was it decided if three (Figs 1, 2) or two (Fig 3) repetitions are needed? Which repetition was included in the analyses? Were all repetitions sequenced? If yes, could these all be analysed? Perhaps the read counts could be normalised based on the control (inoculum) read counts.

Further questions, comments and recommended edits are included in the attached, downloadable PDF file.

Reviewer 2 Report

Comments and Suggestions for Authors

This study describes the use of Nanopore amplicon sequencing to evaluate the fitness effects of specific mutations in a competition assay.

General comments

It's suggested that this procedure will allow lab researchers with limited facilities to conduct competition assays. However, the authors gloss over the need to create viruses that only differ by the mutation under consideration and have a highly restricted quasispecies. The authors also fail to describe how the read data is processed, analyzed and interpreted.

Specific comments

Materials and Methods

Line 103 - How many PCR cycles were used?

Table 1 - This table is confusing the way it is formatted. It is difficult to tell where one primer sequence ends and the next begins. The column title "Genotype of interest" does not accurately describe the information in that column. At this point, the text has provided no information about what a k-mer is. For ease of understanding, k-mer could be relabelled as "sequence". The information about k-mers and Unix "grep" does not belong in the table caption. There should be a section about how sequence reads are analyzed that this information belongs in.

Line 113 - Where is the sequencing service located? What library prep was done and what was returned for analysis?

Figure S2 was unreadable.

How many range of total reads were returned for each sample? What percentage of genomes does this equate to? How were PCR replicates handled? The k-mers were obviously larger than the individual mutation as least for nsp3 N24A. How were non-identical nucleotides outside the nutation of interest interpreted? How were non-identical nucleotides in the mutation of interest interpreted?

Results

Section 3.1 is unnecessary as it is a recap rather than results.

Section 3.2 - Since this mutation is known to have a lower fitness than the wildtype as shown in growth curve, why would a competition assay be needed?

If you know that the N24A mutant has a fitness defect wouldn't it be better to make it the higher number of genomes put in the competition assay rather than putting it at a numerical disadvantage as well. If the wildtype outcompetes the mutant when the wildtype has fewer genomes bolsters the interpretation that the mutant is less fit. Perhaps using the wildtype:mutant at 5:1 and 1:5 would be a good way to do it.

Figure 1 - Panel B has no statistical significance indicated. Why is the WT not on the graphs? It's understood that the mutant is the genotype of interest, but the fitness disadvantage should be against the WT at 24 hrs not the inoculum.

Line 167 - Do you mean 6ng/ul after RT-PCR? or is this total RNA? 6ng of viral RNA before RT-PCR amplification sounds like a huge amount.

Figure 2 - Where does the virus come from at -8 hours? At this time, wouldn't the cells be uninfected?

Figure 3 - Panel A does not have statistical significance indicated.

Discussion - There needs to be more discussion about how Nanopore sequencing is error-prone and how this affects the reliability of read sequences.  In addition, if the read depth is between 50 and 100 for the target region, how many of these are PCR replicates rather than individual genomes being sequenced?

Comments on the Quality of English Language

The English is satisfactory.
